# Out-of-Dynamics Imitation Learning from Multimodal Demonstrations

**Yiwen Qiu**[1], **Jialong Wu**[2], **Zhangjie Cao**[3], **Mingsheng Long**[2✉]

[1]Department of Automation, Tsinghua University, China
[2]School of Software, BNRist, Tsinghua University, China
[3]Department of Computer Science, Stanford University, Stanford, CA 94305, USA
`{qywmei,wujialong0229}@gmail.com`
`caozj@cs.stanford.edu, mingsheng@tsinghua.edu.cn`

**Abstract:** Existing imitation learning works mainly assume that the demonstrator who collects demonstrations shares the same dynamics as the imitator. However, the assumption limits the usage of imitation learning, especially when collecting demonstrations for the imitator is difficult. In this paper, we study out-of-dynamics imitation learning (OOD-IL), which relaxes the assumption to that the demonstrator and the imitator have the same state spaces but could have different action spaces and dynamics. OOD-IL enables imitation learning to utilize demonstrations from a wide range of demonstrators but introduces a new challenge: some demonstrations cannot be achieved by the imitator due to the different dynamics. Prior works try to filter out such demonstrations by feasibility measurements, but ignore the fact that the demonstrations exhibit a multimodal distribution since the different demonstrators may take different policies in different dynamics, which hinders learning an accurate measurement. We develop a better transferability measurement to tackle this newly-emerged challenge. We first design a novel sequence-based contrastive clustering algorithm to cluster demonstrations from the same mode to avoid the mutual interference of demonstrations from different modes and then learn the transferability of each demonstration with an adversarial-learning based algorithm in each cluster. Experiment results on several MuJoCo environments, a driving environment and a simulated robot environment show that the proposed transferability measurement more accurately finds and down-weights non-transferable demonstrations and outperforms prior works on the final imitation learning performance. We show the videos of our experiment results on our website.

**Keywords:** Imitation Learning, Out-of-Dynamics Imitation Learning

## 1   Introduction

Imitation learning is a widely-used policy learning paradigm to solve robotics and control tasks, which learns the policy from demonstrations [1, 2]. Standard imitation learning assumes that the demonstrator who collects the demonstrations shares the same dynamics with the imitator, which means that their state spaces, action spaces and the transition models are all the same [3, 4, 5]. However, such strict assumption limits the practical usage of imitation learning, especially when it is difficult to collect demonstrations in the imitator's environment.

In this paper, we relax the assumption to that the demonstrator and the imitator share the same state space but the dynamics could be different, i.e., their action spaces and the transition models could be different. We name the new imitation learning setting as out-of-dynamics imitation learning (OOD-IL). The new assumption poses fewer requirements on the demonstrator and enables imitation learning to utilize a broader range of demonstrations. For example, in autonomous driving, to learn the policy of an autonomous vehicle in a new city, we may use the database of driving behavior of human vehicles containing useful information instead of manually collecting demonstrations on the autonomous vehicle. But such a database cannot be used by standard imitation learning since the vehicles have different dynamics and the city environments can be different. Furthermore, in real

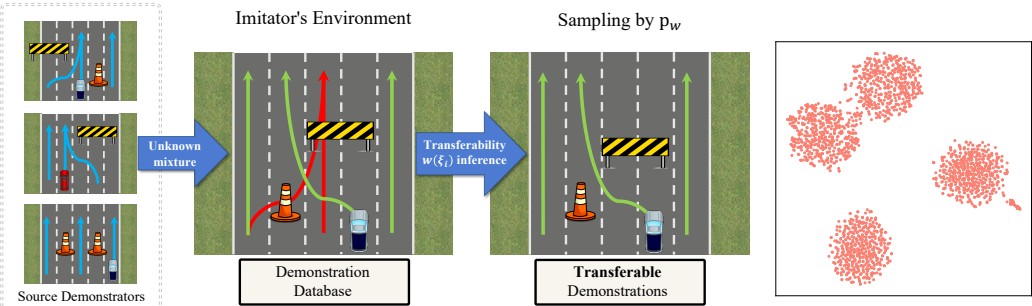

(a) Problem setting and the general framework.  (b) Multimodal distribution.

Figure 1: (a) We are given demonstrations collected from an unknown mixture of demonstrators, which exhibit a multimodal distribution since different demonstrators may take different policies. Some demonstrations (red) may not be achievable by the target agent while others (green) are achievable. Learning from these unachievable demonstrations introduces unknown behaviors. We develop a transferability measurement to mitigate negative impact of these non-transferable demonstrations and learn from transferable ones. (b) A t-SNE visualization of demonstrations collected from 4 dynamics in the MuJoCo environment.

applications, it is often difficult and expensive to obtain enough data from a single source, thus it is beneficial to utilize data from a mixture of massive data sources for better real-world performance.

As shown in Figure 1(a), OOD-IL introduces a challenge that the mixture of demonstrations collected in different dynamics may not be achievable by the imitator, and thus are non-transferable. The state-of-the-art work constructs feasibility measurement to down-weight the non-transferable demonstrations by learning a unimodal policy from a feasibility-MDP (f-MDP) [6]. However, as shown in Figure 1(b), we notice that the prior work overlooks an important fact: the demonstrations collected from different demonstrators exhibit a multimodal distribution since different demonstrators may take different policies. The multimodal demonstrations make it difficult to learn a unimodal policy from the f-MDP as well as learn an accurate feasibility measurement. Furthermore, f-MDP suffers from slow and inaccurate optimization difficulties due to a step-by-step optimization procedure.

In this paper, we address OOD-IL with the above challenges by developing a better transferability measurement to determine how transferable each demonstration is for the imitator. *To fully remove the interference from multimodal distribution*, we design a sequence-based contrastive clustering algorithm to simultaneously learn a hidden space and cluster the trajectories in the hidden space, which ensures each cluster introduces a unimodal trajectory distribution. Then we learn the transferability for each cluster respectively by generative adversarial imitation learning (GAIL) [5], a much easier-to-optimize objective than f-MDP. The discriminator in GAIL distinguishes transitions from the demonstrators' distribution to the imitators' distribution, which serves as the transferability measurement to indicate the likelihood for a demonstration trajectory to be reproduced by the imitator. Experiment results on several MuJoCo environments, a driving environment, and a simulated Franka Panda Arm environment show that by reweighting demonstrations with the proposed transferability measurement, the final imitation learning policy outperforms all baselines and achieves the state-of-the-art performance.

## 2 Related Works

**Standard Imitation Learning.** Imitation learning learns a policy to imitate the behavior in demonstrations. Existing algorithms can be roughly categorized into three types: Behavior Cloning (BC), Inverse Reinforcement Learning (IRL), and Generative Adversarial Imitation Learning (GAIL). BC utilizes supervised learning to directly learn the policy from state-action pairs [7]. Following works propose dataset aggregation [3] or policy aggregation [8, 9] to address the compounding errors problem in BC. Torabi *et al.* [10] try to learn from state sequences by first recovering the actions through an inverse dynamics model and then conducting behavior cloning. IRL first recovers the reward function from demonstrations and then learns the policy using the learned reward [11, 12, 4, 13]. GAIL-based works match the occupancy measure between the learned policy and the demonstrations through adversarial learning to seek the optimal policy [5, 14]. Also, GAIL is demonstrated to suc-

cessfully imitate state sequences [15, 16, 17]. However, all the methods assume that the demonstrator and the imitator share the same dynamics, which violates the assumption of OOD-IL.

**Out-of-dynamics Imitation Learning.** To address the OOD-IL problem, there are works learning a correspondence model [18] between the demonstrator and the imitator [19, 20, 21, 22, 23]. However, these methods assume that strict correspondence, i.e. a one-to-one mapping between the state spaces and action spaces, exists between the demonstrator and the imitator. This assumption can be violated in real-world applications, e.g., a 7-DoF robot arm and a 3-DoF robot arm have no such correspondence since some behavior of the 7-DoF robot cannot be realized by the 3-DoF robot. Recent works relax the assumption to that the demonstrator and the imitator only share the state space [24, 25]. One line of work aims to maximally follow the demonstrations [24, 26, 27, 28], but following the non-transferable demonstrations is impossible.

Cao *et al.* [25] develop a feasibility measurement to down-weight non-transferable demonstrations by learning an inverse dynamics model, but the learned inverse dynamics may not generalize to all the demonstrations. To address this difficulty, the state-of-the-art work improves the feasibility by learning the optimal policy in f-MDP [6]. However, f-MDP suffers from two limitations. Firstly, f-MDP enforces one unimodal policy to maximally imitate the demonstrations but the demonstrations with a multimodal distribution are hard to be modeled by such a policy. Secondly, at each time step, only when the policy is optimized in all prior time steps can it be optimized to maximize the reward at the current time step. Such inappropriate design makes it difficult and inefficient to learn the optimal policy of f-MDP. We instead use the discriminator in GAIL to learn the transferability measurement.

**Contrastive Clustering.** Unsupervised deep clustering methods with the aid of contrastive representation learning have been proposed for computer vision [29, 30, 31] and natural language processing [32]. To decompose the intrinsic multimodal distribution of demonstrations, we also adopt the intuition of jointly learning representations and clustering in an end-to-end manner. While prior works either focus on visual representations in the form of a single fixed-length vector [30, 31] or rely on data augmentations with a heavy computational burden [32], our Sequence-based Contrastive Clustering is unique in handling trajectory data with various sequential lengths, by utilizing simple and efficient down-sampling to construct positive pairs of contrastive learning.

## 3 Out-of-dynamics Imitation Learning

In OOD-IL, we aim to learn a policy for the agent of interest (the imitator) from demonstrations collected from multiple demonstrators with different dynamics from the imitator. Formally, we model both the demonstrators and the imitator as a standard Markov decision process (MDP) $\mathcal{M} = \langle \mathcal{S}, \mathcal{A}, p, \mathcal{R}, \rho_0, \gamma \rangle$. Here $\mathcal{S}$ is the state space, $\mathcal{A}$ is the action space, $p : \mathcal{S} \times \mathcal{A} \times \mathcal{S} \to [0, 1]$ is the transition probability and $\mathcal{R} : \mathcal{S} \times \mathcal{S} \to \mathbb{R}$ is the reward function, which is solely defined on states and shared between the demonstrators and the imitator. Such design aims to satisfy the basic requirement of imitation learning that the demonstrators and the imitator should finish the same task [24, 25]. $\gamma$ is the shared discount factor. The action spaces $\mathcal{A}$ and the transition probability $p$ of different demonstrators and the imitator are different. In particular, we use $\mathcal{M}^i = \langle \mathcal{S}, \mathcal{A}^i, p^i, \mathcal{R}, \rho_0, \gamma \rangle$ to indicate the MDP of the imitator. A policy for the imitator $\pi^i : \mathcal{S} \times \mathcal{A}^i \to [0, 1]$ defines a probability distribution over the action space in a given state. The policy is evaluated by the expected return, which is defined by $\eta_{\pi^i} = \mathbb{E}_{s_0 \sim \rho_0, \pi^i} \left[ \sum_{t=0}^{\infty} \gamma^t \mathcal{R}(s_t, s_{t+1}) \right]$, where $t$ indicates the time step.

We formalize the policy learning as an imitation learning problem where the reward function $\mathcal{R}$ is unknown. We aim to learn the policy $\pi^i$ from a set of demonstrations collected by different demonstrators $\Xi = \{\xi_1, \xi_2, \dots\}$ where each trajectory is a sequence of states $\xi = \{s_0^d, s_1^d, \dots, s_H^d\}$. Here we use state trajectories since different action spaces make it impossible to imitate actions. Since we put no assumption on the dynamics of the demonstrators, there is no guarantee that an optimal policy could be learned, e.g., in the extreme case, all the demonstrations could be non-transferable and a random policy will be learned. Our goal is to learn a policy with as high return as possible.

Since the dynamics of the demonstrators and the imitator are different, the transitions $(s_t^d, s_{t+1}^d)$ in the demonstrations may not be realizable by the imitator, i.e., no action $a_t^i$ in the imitator's action space $\mathcal{A}^i$ could make $p^i(s_t^d, a_t^i, s_{t+1}^d) > 0$. Such demonstrations provide no useful information for the imitator and are non-transferable. However, given such a set of demonstrations from a mixture of demonstrators, it is non-trivial to directly test whether a trajectory is achievable by reproducing the trajectory in the target environment since there is no action in demonstrations. Furthermore, different

demonstrators may take different policies due to different dynamics and may take different actions even at the same state[1]. This multimodal distribution of the demonstrations makes prior works on transferability measurement ineffective. In this paper, we aim to remove the interference from the multimodal distribution of the demonstrations and learn transferability to mitigate the negative impact of non-transferable demonstrations.

## 3.1 Sequence-based Contrastive Clustering

Since directly imitating from a mixture of demonstrations will make a unimodal policy suffer, we need to fully capture and decompose the intrinsic multimodal distribution of data collected from different demonstrators. Thus, we cluster demonstrations before learning their transferability to make sure they belong to a unimodal distribution. Motivated by contrastive learning [33], we learn a sequence feature extractor and a distance metric in the sequence feature space by contrastive learning over carefully-designed positive and negative pairs. We construct positive pairs by randomly subsampling fixed-length sub-trajectories from the same demonstration and treat sub-trajectories of different demonstrations as negative pairs. Such a design satisfies our requirement since the positive pairs are from the same trajectory, and thus are from the same mode. At each batch, we first sample $N$ trajectories from $\Xi$ and for each trajectory $\xi_i$, we take two sub-trajectories $\xi_{2i-1}^{\text{sub}}$ and $\xi_{2i}^{\text{sub}}$. We then derive the contrastive learning loss for this batch of $2N$ sub-trajectories as follows:

$$
\mathcal{L}_{\text{contrast}} = -\frac{1}{N} \sum_{i=1}^{N} \log \frac{\exp(\langle F(\xi_{2i-1}^{\text{sub}}), F(\xi_{2i}^{\text{sub}}) \rangle)}{\sum_{i' \neq 2i-1, i' \neq 2i} \exp(\langle F(\xi_{2i-1}^{\text{sub}}), F(\xi_{i'}^{\text{sub}}) \rangle) + \exp(\langle F(\xi_{2i-1}^{\text{sub}}), F(\xi_{2i}^{\text{sub}}) \rangle)},
\tag{1}
$$

where $F$ is the feature extractor modeled as a recurrent neural network and $\langle \cdot, \cdot \rangle$ indicates the cosine distance. The loss maximizes the cosine similarity of features between positive pairs.

We embed contrastive learning into clustering to simultaneously ensure that the positive pairs are close and the clustering structure is learned. We first initialize a matrix $\mathbf{C}$ of size $\dim(F) \times K$, where $\dim(F)$ is the dimension of the output of $F$ and each column $\mathbf{c}_k$ of $\mathbf{C}$ represents the center of cluster $k$, with $K$ clusters in total. For each input sub-trajectory $\xi_n^{\text{sub}}$, we assign a one-hot cluster label $\mathbf{y}_n \in \{0, 1\}^K$ as follows:

$$
y_{n,j} = \begin{cases} 1, & j = \arg\min_{k=1,\cdots,K} \|F(\xi_n^{\text{sub}}) - \mathbf{c}_k\|_2 \\ 0, & \text{otherwise} \end{cases}
\tag{2}
$$

where we assign the cluster label with the nearest cluster center to a sub-trajectory. Then we introduce the clustering objective with the contrastive learning loss $\mathcal{L}_{\text{contrast}}$ embedded into it:

$$
\mathcal{L}_{\text{cluster}} = \mathcal{L}_{\text{contrast}} + \frac{\lambda}{2} \|F(\xi_n^{\text{sub}}) - \mathbf{C}\mathbf{y}_n\|_2^2.
\tag{3}
$$

Every time we update the feature extractor $F$, we reassign the cluster label of each sub-trajectory by Eqn. (2) and update the cluster center by the following process:

$$
\mathbf{c}_k \leftarrow (1 - \beta)\mathbf{c}_k + \beta \sum_{n=1}^{2N} \mathbb{I}[y_{n,k} = 1]F(\xi_n^{\text{sub}}),
\tag{4}
$$

where $\mathbb{I}$ is an indicator taking 1 only when the condition is satisfied and 0 otherwise, and $\beta = \frac{1}{\sum_{n=1}^{2N} \mathbb{I}[y_{n,k}=1]}$ automatically controls the learning rate. We simultaneously optimize the contrastive learning objective and the clustering objective to encourage them to benefit from each other, where the cluster structure is learned to keep trajectories from the same mode to be in the same cluster. Our contrastive clustering algorithm derives $K$ clusters of trajectories $\Xi_k|_{k=1}^K$, where $\Xi_k$ contains trajectories with cluster label $k$. We show the algorithm procedure in Appendix.

## 3.2 Adversarial Transferability Measurement

Given that the sequence-based contrastive clustering algorithm removes the mutual interference of demonstrations from different modes, we then learn the transferability within each cluster. We

---

[1]This work mainly focuses on the case where the dynamics are nearly-deterministic like robotic applications so unimodal policies can rarely realize multimodal demonstrations.

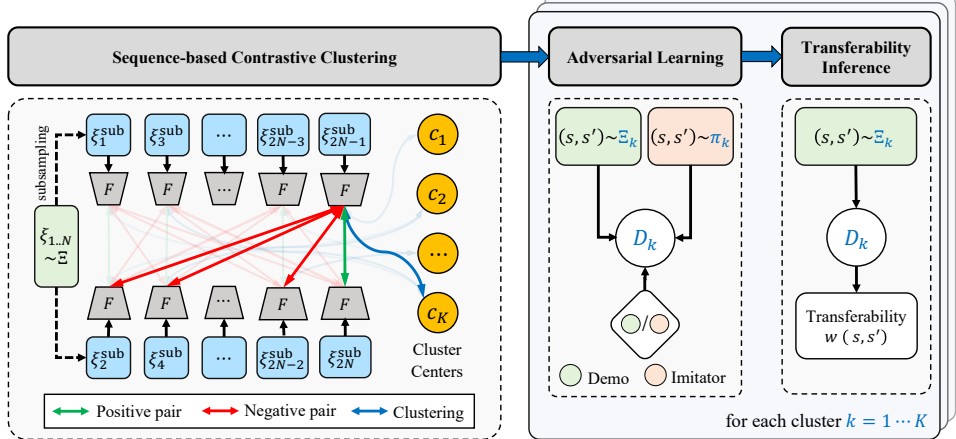

Figure 2: The outline of our whole algorithm can be divided into two phases. The first phase is sequence-based contrastive clustering where we simultaneously conduct contrastive learning and clustering. We create positive pairs by subsampling different sub-trajectories from the same trajectory and use sub-trajectories from different trajectories as negative pairs. The second phase is learning transferability where we conduct an adversarial-learning based algorithm in each cluster.

design a new transferability measurement by an adversarial-learning algorithm based on generative adversarial imitation learning (GAIL) [5]. Our key insight is that the GAIL discriminator output indicates the likelihood for a state transition to either come from the demonstration distribution or the policy distribution. We train a GAIL policy $\pi_k$ and a GAIL discriminator $D_k$ for each cluster of trajectories. The loss is defined as follows:

$$\mathcal{L}_{\text{tran}} = -\sum_{k=1}^{K} \left( \mathbb{E}_{(s_t^d, s_{t+1}^d) \sim \Xi_k} \log(1 - D_k(s_t^d, s_{t+1}^d)) + \mathbb{E}_{(s_t^{\pi_k}, s_{t+1}^{\pi_k}) \sim \pi_k} \log(D_k(s_t^{\pi_k}, s_{t+1}^{\pi_k})) \right). \quad (5)$$

Here $\pi_k$ indicates the current policy in training and $D_k$ indicates the discriminator. We use label $0$ for the state transitions in the demonstrations and $1$ for the state transitions collected from the policy.

After convergence, the discriminator for each mode outputs a value in $[0, 1]$ reflecting how likely an input transition is drawn from the state transition distribution derived by the imitator's policy. Thus for a state transition in the demonstrations, if it has a discriminator output close to $1$, it is more possible to be drawn from the imitator's policy and achievable by the imitator. Thus, we quantify the transferability for a state transition as follows:

$$w(s_t^d, s_{t+1}^d) = \sum_{k=1}^{K} \mathbb{I} \left[ (s_t^d, s_{t+1}^d) \in \Xi_k \right] D_k(s_t^d, s_{t+1}^d). \quad (6)$$

Note that the discriminator is strengthened by the adversarial training paradigm and thus has a strong ability to discriminate whether a transition is from the demonstrations or the imitator's policy. The optimization of the discriminator is demonstrated to be efficient and effective [5], which requires no time-consuming step-by-step optimization in f-MDP [6].

**Transferablity-sampling Imitation Learning.** We finally embed the transferability quantified by Eqn. (6) into imitation learning. We normalize the transferability of all the state transitions into a sampling distribution, where state transitions with larger transferability will be sampled more often:

$$p_w(s_t^d, s_{t+1}^d) = \frac{w(s_t^d, s_{t+1}^d)}{\sum_{(s_{t'}^d, s_{t'+1}^d) \in \Xi} w(s_{t'}^d, s_{t'+1}^d)}. \quad (7)$$

Using the fixed sampling distribution $p_w$, we can embed our transferability into any imitation-learning-from-observations algorithm [10, 5, 14]. We use GAIL [5] here to train the final policy:

$$\mathcal{L}_{\text{GAIL}} = -\mathbb{E}_{(s_t^d, s_{t+1}^d) \sim p_w} \log(1 - D(s_t^d, s_{t+1}^d)) - \mathbb{E}_{(s_t^\pi, s_{t+1}^\pi) \sim \pi} \log(D(s_t^\pi, s_{t+1}^\pi)). \quad (8)$$

## 4 Experiments

We experiment with three MuJoCo environments, a simulated Driving environment, and a simulated Franka Panda Arm environment. We compare our approach with a standard imitation learning algorithm: GAIL [5], imitation learning with a measure of feasibility: ID [25] and f-MDP [6]. The original ID learns the inverse dynamics model from random trajectories far from demonstrations and thus makes the learned inverse dynamics not work well on demonstrations. We create an advanced ID baseline by learning a GAIL policy to generate trajectories as the data for inverse dynamic learning. Such trajectories are closer to demonstrations and make the inverse dynamics work better on demonstrations. We call the original ID as ID-Random and the advanced ID as ID-GAIL.

We further conduct analyses including an ablation study to verify the efficacy of each component and include a visualization of the learned transferability, the results for different compositions of demonstrations, and the performance gain when we are given a larger budget to collect demonstrations in the Appendix. Code is available at `https://github.com/EvieQ01/OODIL`.

### 4.1 MuJoCo

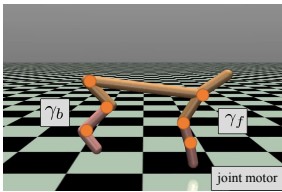 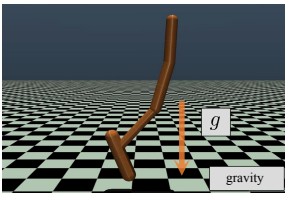 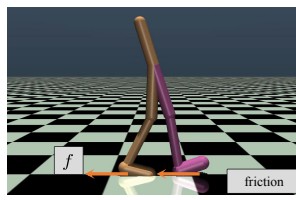

Figure 3: The illustration of the varying dynamics for HalfCheetah, Hopper, and Walker2d.

**Environments.** We illustrate the environments in Figure 3. The **HalfCheetah** is an agent with two legs and a body. We create different dynamics by discounting the force of the front leg and the back leg with two factors $\gamma_f$ and $\gamma_b$ respectively. We create a mixture of demonstrators by setting $(\gamma_f, \gamma_b)$ as (i) $(1, 0.9)$, (ii) $(0.9, 1)$, (iii) $(1, 0.05)$, (iv) $(0.05, 1)$. The imitator has the original force: $(\gamma_f, \gamma_b)$ as $(1, 1)$. The **Hopper** is an agent with one leg consisting of 3 joints. We create different dynamics by varying the gravitational constant. We create a mixture of demonstrators by setting the gravitational constant as (i) $15.0$, (ii) $9.8$, (iii) $2.0$, (iv) $1.0$. For the imitator, we use a gravitational constant of $12.0$. The **Walker2d** is an agent with two legs where each leg consists of 3 joints. We create different dynamics by using different frictions for the feet, i.e., the link that touches the ground. We create a mixture of demonstrators by setting the friction as (i) $24.8$, (ii) $9.9$, (iii) $3.9$, (iv) $1.1$. For the imitator, we use a friction of $19.9$. For all three MuJoCo environments, we follow prior works [6, 25] to collect fewer demonstrations from more transferable environments. We collect $100, 100, 250, 250$ demonstrations respectively for four dynamics in HalfCheetah, $10, 10, 250, 250$ for Hopper and $25, 50, 50, 50$ for Walker2d with 1000 interaction steps per demonstration. We train expert agents with Trust Region Policy Optimization (TRPO) [34] to generate demonstrations.

**Results.** We show the expected return w.r.t. the number of interaction steps for the three environments in Figure 4. In all three environments, we observe that the proposed method achieves the highest return. The baselines even show lower performance than naive GAIL under such multimodal distribution of source demonstrations because f-MDP learns a unimodal policy from the multimodal trajectory distribution, which cannot realize the demonstration trajectories, while ID fails to learn an accurate inverse dynamics model from random trajectories. The results demonstrate the importance of clustering trajectories of the same mode. By learning the transferability in each cluster, we can accurately filter out non-transferable demonstrations and learn from transferable demonstrations.

### 4.2 Driving

**Environment.** In the driving environment, we can easily decide and interpret whether the target car can reproduce the route in the demonstrations. As shown in Figure 5(a), we create a task where a car drives starting from anywhere at the bottom side and ends at the top side. Two obstacles are set with the center at $\frac{1}{4}$ width and $\frac{3}{4}$ width respectively. We create different dynamics by setting obstacles with different widths and setting different speeds for the car, which simulates a realistic scenario where different car models are driving at different places. The reward function is defined as $-1$ for each interaction step, $+1000$ for reaching the goal, and $-1000$ for hitting the obstacle. In this environment, the different lengths and paths of the demonstration trajectories introduce a clear

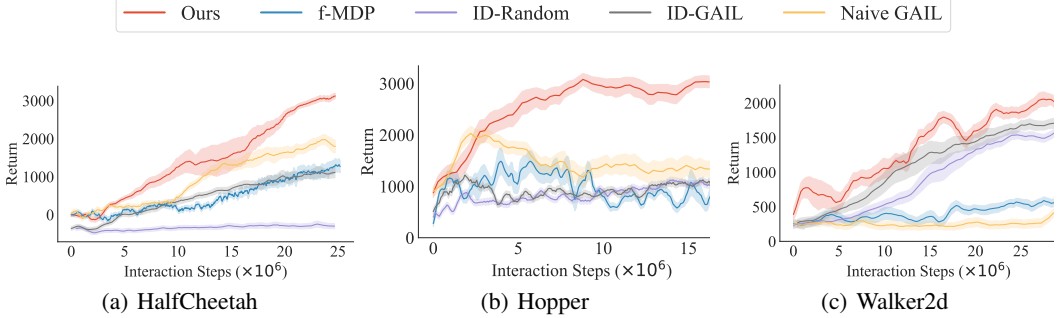

Figure 4: The imitation learning performance of different methods in the three MuJoCo environments.

multimodal distribution, which can demonstrate the importance of clustering trajectories into an accurate mode. We create three demonstrators by setting the obstacles width as $[0.1, 0.5]$, $[0.5, 0.25]$ and $[0.25, 0.25]$ and setting the speed as $1.0$, $1.0$ and $5.0$ respectively. For the target environment, we set the obstacle width as $[0.4, 0.25]$ and the speed as $1.0$. We collect $5 \times 10^5$, $2.5 \times 10^6$, and $2.5 \times 10^6$ interaction steps of demonstrations from each source dynamics, by handmade rules.

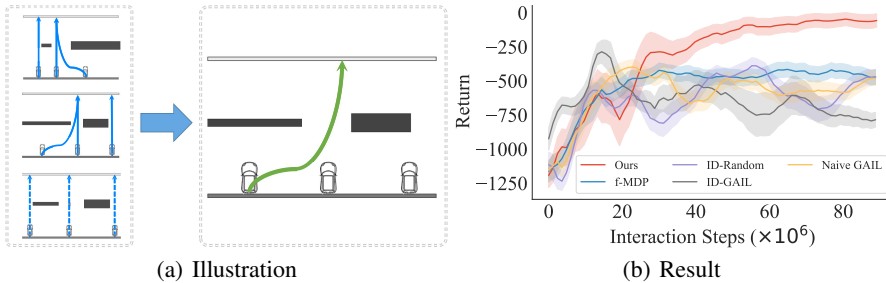

Figure 5: (a) The illustration of the driving environment, showing expert trajectories in multiple source dynamics and the target environment. (b) The results in the driving environment.

**Results.** As is shown in Figure 5(b), we observe that the proposed method outperforms all other baselines. Though ID performs well at the first few interaction steps but the performance drops much then, which can be explained by that the feasibility learned by ID could find good demonstrations in the first few batches of data but makes errors then. That means ID can only learn partially correct feasibility. Instead, our method learns accurate transferability to filter out non-transferable demonstrations and converges stably to a high return. f-MDP, as the state-of-the-art method for filtering non-transferable demonstrations, could not work well on this multimodal distribution of demonstrations, further indicating the importance of clustering trajectories into the correct mode.

### 4.3 Simulated Franka Panda

**Environment.** The environment simulates the Franka Panda Robot Arm[2] with 7 degrees of freedom (DoF), which is implemented in the PyBullet [35]. We create a task of pushing a box from one side of the desk to the other. With the box set at a base position and the robot arm set at a random position with Gaussian distribution, we set the target at the right of the desk. We create different dynamics by disabling different joints of the Robot arm. As shown in Figure 6(a), we create three demonstrators by disabling the No. 1, 4, and 6 joint respectively while disabling the No. 1 and 3 joints for the target imitator. The environment aims to address a real problem to leverage historical data on any robot to learn a policy for a new robot. Similar to the MuJoCo environment, we import more demonstrations from the more dissimilar demonstrator, where the number of interaction steps is $1 \times 10^6$, $1 \times 10^5$, and $1 \times 10^6$ for the environment disabling No. 1, 4, and 6 respectively. The reward function is defined as the current distance to the starting point, and an extra $+1000$ for the box reaching the other side of the table and $-1000$ for the box dropping to the ground or the robot going past the box to the other side of the desk. We make demonstrations by handmade rules.

**Results.** The expected return w.r.t. the number of interaction steps is shown in Figure 6(b). Our proposed method outperforms all the baselines by a large margin. Directly applying GAIL introduces

---

[2]https://www.franka.de/

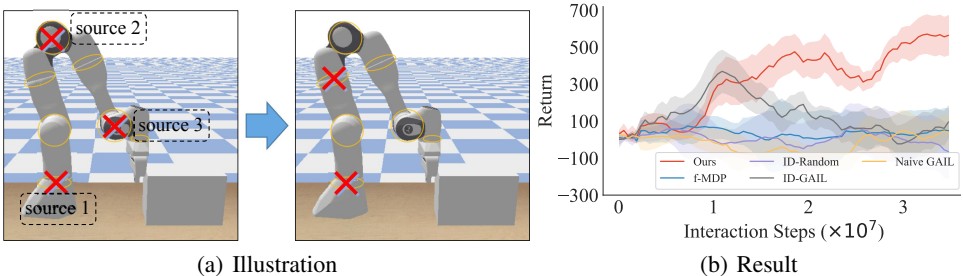

(a) Illustration             (b) Result

Figure 6: (a) The illustration of Simulated Franka Panda environment, where we create different demonstrators by disabling different single joints and disabling two joints for the imitator. (b) The return of different methods in the Simulated Franka Panda environment.

a low performance, which indicates that the demonstrations from other robots cannot be directly utilized to learn a new robot. The experiments show that the proposed method can serve as a data cleaning step to clean the dataset from other robots for a real-robot transfer learning problem.

## 4.4 Analysis

**Ablation Study.** To verify that both our contrastive clustering algorithm and the adversarial-learning based algorithm contribute to the final performance, we compare the performance of our method with its variants by removing the clustering step and learning the transferability directly from the whole set of demonstrations (Ours w/o Cluster), and removing both clustering and the transferability (Ours w/o Cluster, Tran), which directly imitates the whole set of demonstrations. We conduct these experiments in the Driving environment.

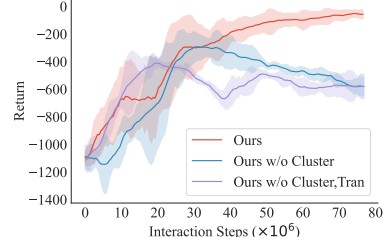

Figure 7: The ablation study on two variants of our method.

The results are shown in Figure 7. We observe that Ours outperforms Ours w/o Cluster, which demonstrates that clustering trajectories within the same mode is important and our contrastive clustering algorithm achieves this goal. Ours w/o Cluster outperforms Ours w/o Cluster, Tran, which demonstrates that transferability is important to filter out non-transferable demonstrations and learn from more transferable ones.

## 5  Conclusion

We propose a new approach to address out-of-dynamics imitation learning (OOD-IL). Noticing that the demonstrations exhibit a multimodal distribution, we propose a sequence-based contrastive clustering algorithm to make trajectories from the same mode fall into the same cluster. We then propose an adversarial-learning based algorithm to learn the transferability of each cluster with the discriminator output. Experimental results on three MuJoCo environments, a driving environment, and a simulated robot environment show that the proposed method can learn a transferability measure to accurately filter out non-transferable demonstrations and learn from more transferable ones.

**Limitations and future work.** While our work has substantially advanced OOD-IL, we believe that this problem merits further study. One primary limitation of our current method is that our method may not afford the computational cost of transferability learning when the number of clusters increases to hundreds or thousands. Pre-training or meta-training [36] before transferability learning may help boost learning efficiency, which is left for future work. A limitation of deploying our method into a real system is that it may not respect safety constraints due to exploration during both transferability learning and imitation learning. Although this challenge is common for real systems, out-of-dynamics learning may exacerbate it and safe policy learning techniques can be adopted. Another promising future direction is to exploit a wider variety of demonstrations, thus enabling large-scale imitation learning. Although our work relaxes the stringent assumption of identical dynamics, it is also limited since we assumed that collected demonstrations are optimal in the target environment, while it might not hold true in varying dynamics. In the future, we plan to address this combined challenge of multimodal OOD-IL and learning from sub-optimal demonstrations [37].

## Acknowledgments

We would like to acknowledge the support of our wonderful coworkers: Yang Shu, Baixu Chen, Haixu Wu, and Yipeng Huang, whose generous help is an integral part of this work. This work was supported by the National Key Research and Development Plan (2020AAA0109201), National Natural Science Foundation of China (62022050 and 62021002), Beijing Nova Program (Z201100006820041), and BNRist Innovation Fund (BNR2021RC01002).

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
