# OpenReview forum: "Out-of-Dynamics Imitation Learning from Multimodal Demonstrations"
_robot-learning.org/CoRL/2022/Conference — CoRL 2022 Poster_

### Official Review · Reviewer_UJgv · 2022-07-28

**Originality:** Very Good
**Technical Quality:** Very Good
**Clarity Of Presentation:** Very Good
**Impact:** 4

**Recommendation:**

Strong Accept: I recommend accepting the paper and will argue for my recommendation even if other reviewers hold a different opinion.

**Summary:**

This paper proposed a novel imitation learning method that utilizes multimodal demonstrations from different dynamics and action spaces.
The main contribution of the proposed method is to develop an algorithm that relaxes the assumptions in the general imitation learning and explicitly takes into account the multimodal nature of demonstrations, which has been overlooked in previous studies.

**Issues:**

I would like to see additional ablation studies for checking the effect of the number of clusters.
In particular, please clarify the limitation in terms of learning efficiency and computational cost.
In addition, we would like to know if the Interaction Steps on the horizontal axis of the learning curve in the paper include those required for learning pi_k.
If not, the authors should clarify the number of iteration steps required for learning pi_k, which reveals the decrease of learning efficiency in total.

From the benchmarks provided, it seems that the sub dataset extracted according to the transferability measurement is designed to avoid multimodality.
It would be better to evaluate the performance when it is multimodal, if possible.

Only for defining transferability measurement, it may not be necessary to cluster the dataset, and it seems sufficient to construct a metric space.
It would be good if the authors could add the necessity of clustering in this regard.

**Quality Of The Limitations Section:**

Additional details required

**Reviewer Expertise:**

3: The reviewer is fairly confident that the evaluation is correct

**Robotics Focus:**

Highly relevant to robotics but no hardware experiments

**Strengths And Weaknesses:**

Strengths:
- Under a more general problem setting, this paper succeeded in imitating from the multimodal demonstrations without simlifying the problem.
- The proposed method is able to absorb differences not only in dynamics but also in the action space.
- The performance of the proposed method was properly verified by benchmarks, including ablation studies.

Weaknesses:
- When dealing with multimodality, it is inconvenient to design the number of clusters K to be specified in advance.
In particular, since the proposed method requires training pi_k for each cluster, it is not desirable to guarantee generality by making K too large without consideration, but too small K cannot capture the multimodality.
- I think the two-stage GAIL should eventually make the learning process highly unstable.
- The final pi to be acquired is obtained by GAIL using a subset of the multimodal dataset that is close to the imitator's dynamics, which should inherit multimodality.
In that case, depending on the probability model of pi, it may fail to capture the multimodality and underfitting may occur.

**Summary Of Recommendation:**

The problem setting that this paper challenges is very important and versatile from the viewpoint of practicality, and the proposed method steadily resolved the issues that arise in this, although its approach seems to be heuristically constructed.
The performance of the proposed method was properly verified, and the effectiveness of the method seems to be confirmed.

---

> ### Author Response · Authors · 2022-08-24
> **Response to Reviewer Ujgv (Part 1)**
>
> We would like to sincerely thank Reviewer Ujgv for providing the detailed review and the positive evaluation of the quality of our paper.
>
> **Q1: The design of the number of cluster K and ablation studies of it? In addition, we would like to know if the Interaction Steps on the horizontal axis of the learning curve in the paper include those required for learning** $\pi _k$.
>
> **(1)** We appreciate the reviewer's question about the design of K as it can be hard to determine in advance. K is unknown to us but we can estimate it empirically, for example by visualizing trajectories through t-SNE. We have conducted an ablation study of K in the $\underline{\text{Section C.1 of supplementary material}}$, and the result shows that the final performance isn't very sensitive to K and an approximation of K is generally enough. The reviewer can refer to our response for **Q6 of Reviewer Zz2p** for further explanation.
>
> **(2)** Thank you for reminding us to include the interaction steps to learn $\pi_k$ to specify the overall computational cost. The followings are the overall count of interaction steps for each algorithm, while numbers separated by '$+$' indicate different stages of the overall algorithm. f-MDP and ID each contain a stage for transferability (feasibility) learning and a stage for imitation learning as well as Ours. And ID-GAIL additionally requires a stage for imitation learning at the beginning.
>
> |   Naive GAIL    |              f-MDP              |              ID               |                  ID-GAIL                  |               Ours                |
> | :-------------: | :-----------------------------: | :---------------------------: | :---------------------------------------: | :-------------------------------: |
> | $1.5\times10^7$ | $1\times10^6 + 1.5\times10^7$ | $1\times10^4 + 1.5\times10^7$ | $1\times10^7 +1\times10^4+ 1.5\times10^7$ | $\lt K\times10^7 + 1.5\times10^7$ |
>
> We use a fixed number of interaction steps for all algorithms in the final imitation ($1.5\times 10^7$). **Each algorithm other than naive GAIL requires extra interaction steps to compute the transferability (feasibility) measurement**. Compared with other baselines, we indeed increase the interaction steps while obtaining more than 300% increase in the final performance in Hopper environment, 60% increase in the final performance in HalfCheetah environment. Note that $K\times10^7$ is a rough upper bound of interaction steps for our transferability learning on all clusters altogether. Since we have fewer demonstrations within each cluster compared to the whole dataset, it is often easier to be optimized and faster to converge. Sometimes, this stage can be done in much fewer steps than $K \times 10^7$. Moreover, the transferability with $K$ clusters can be learned parallelly and the time for this step may further decrease. We have added discussion on the limitation of our method regarding learning efficiency in the revised paper in Section 5 of main paper.

---

> > ### Author Response · Authors · 2022-08-24
> > **Response to Reviewer Ujgv (Part 2)**
> >
> > **Q2**: **The effect of multimodal after transferring and the necessity for clustering.**
> >
> > **(1)** We appreciate that you proposed we should consider the effect of multimodality in transferable demonstrations after clustering and transferring. However, we would like to note that it is not within our main problem scope to deal with a multi-modal policy learning issue. The goal of our design is to **mitigate the negative impact of multi-modal distribution on transferability learning** and encourage positive transfer.  As we mentioned in response to **Q3 of Reviewer Zz2p**, we can **implement the policy by GMM instead of the original uni-modal Gaussian policy but it is faced with several optimization problems**. For example, the KL divergence of two GMM policies used in the TRPO/PPO step does not have a closed-form solution, but we can only get a very loose bound. The problem that the reviewer has mentioned might be another interesting field to be studied, but **the main concern here is that the multi-modality of demonstrations will cause transferability metric learning to fail**. In addressing this issue, our algorithm turns out to be effective.
> >
> > **(2)** Further, if we make the problem clear, we can see the necessity for clustering. Here **multimodality has a negative impact on 2 things: the final performance for imitation learning and the measurement for transferability**. The latter is a crucial component of the first one. The clustering step serves to derive an accurate transferability metric, without which we cannot mitigate the mutual influence of multimodality for such metric learning. After removing the clustering step, final performance drops due to failure to capture an accurate transferability as our ablation study in the $\underline{\text{Section 4.4 of main paper}}$ shows:
> >
> > |                     | Baseline (w/o Cluster, w/o Tran) | Ours w/o Cluster |     Ours      |
> > | :-----------------: | :------------------------------: | :--------------: | :-----------: |
> > | Driving-multi-modal |            -576 (±49)            |    -494 (±68)    | **-71** (±66) |
> >
> > We observe that 'Ours' outperforms 'Ours w/o Cluster', which demonstrates that **clustering trajectories within the same mode is important and our contrastive clustering algorithm achieves this goal**.

---

> > > ### Comment · Reviewer_UJgv · 2022-08-27
> > > **Thank you for your response**
> > >
> > > Thank you for responding to my comment.
> > > Regarding the last response about the need for clustering, the results seem to indicate that the metric space itself is not sufficiently separated, and therefore, clustering would make the ambiguous boundaries between other clusters clear.

---

> > > > ### Author Response · Authors · 2022-08-27
> > > > **Thanks for the Response of Reviewer UJgv**
> > > >
> > > >  We'd like to thank you again for your time and efforts in providing a valuable review and carefully judging our feedback. We are glad that we have addressed your concerns and improved our work with your help.

---

### Official Review · Reviewer_Zz2p · 2022-07-29

**Originality:** Good
**Technical Quality:** Good
**Clarity Of Presentation:** Good
**Impact:** 3

**Recommendation:**

Weak Accept: I recommend accepting the paper, but will not argue for my recommendation if the majority of other reviewers have a different opinion.

**Summary:**

This paper focuses on recognizing multi-modalities in the learning from demonstrations, where the demonstrations come from different demonstrators, system dynamics, environments, etc. Building a demonstration clustering and transferability measuring procedures on top of the imitation learning allows the model to evaluate the similarity of the current and the demonstrated states.

**Issues:**

- A question in my mind, what is the reason for not using a mixture model, e.g. GMM, to model this multi-modality demonstrations behavior?
- I like Figure 2 in the appendix, but I would like to know how the authors compute the transferability of the other 3 methods.
- As I said in the weakness section, what is the bottleneck of getting a real robot experiment? From my understanding, a task like "driving" should be able to be achieved by a robot in task space. In general, it should be easier for imitation learning to work with a real system than RL, especially because the authors presented a lot of real system settings in the introduction section.
- Where did the demonstrations come from? From RL agent or some human input or hard-coded trajectories?
- The number of clusters, K, seems crucial in the current model. So, what is the relationship between K and the number of demonstrators? Imagine a practical usage case, like autonomous driving, with different cars, weather, and road conditions, even rules (driving on the left in the UK vs. on the right in the US), it seems that K will be very big. So what is the scalability of the current model?


**Quality Of The Limitations Section:**

Additional details required

**Reviewer Expertise:**

3: The reviewer is fairly confident that the evaluation is correct

**Robotics Focus:**

Highly relevant to robotics but no hardware experiments

**Strengths And Weaknesses:**

Pros:

- The motivation of the paper is well explained, i.e. help distinguish demonstrations from different dynamics and therefore allows the imitator to imitate the demos sharing similar states transitions without interference.
- The three parts of the model, i.e. clustering demos, measuring transfer ability, and training the imitation learning model with transferability are well presented.
- The paper has a proper ablation study and further Hyper-parameter tuning details in the appendix.


Cons:
- It is a pity that no real-robot experiment is conducted, which decreases my confidence of mine that this work can be easily deployed in a real system. Maybe the authors can share some knowledge or experience about the limitations of deploying it in a real system.
- The experiment details of the third task is a little bit unclear to me, i.e., what is this "moving one box from one side to the other"? The task description and Fig.6 do not give me enough information about the experiment, and neither does the supplementary material.



**Summary Of Recommendation:**

In general, I think this paper reaches the minimum threshold of CORL, as it is well-written, especially in the model structure part. The reasons that I do not feel very confident to recommend this paper are (1) I do not have strong background knowledge of the methods used in this paper or related works and (2) there is no real robot experiment, which is not common in imitation learning. I am looking forward to getting some feedback from the authors regarding the reasons for not deploying current work in a real system and the scalability of the number of demonstrators.

---

> ### Author Response · Authors · 2022-08-24
> **Response to Reviewer Zz2p (Part 1)**
>
> Many thanks to Reviewer Zz2p for providing a thorough review and valuable suggestions.
>
> **Q1: No real-robot experiments.**
>
> We apologize for the lack of real-robot experiments. Due to the COVID-19 pandemic, the robot in our lab is inaccessible to us, so we regret that we cannot conduct a real-robot experiment to verify the effectiveness of our method.
>
> Following the suggestion of the reviewer, we have enriched the limitation part of our paper, discussing safety considerations when deploying our method into a real system, in Section 5 of the revised paper. Please check our revised paper for this updated part.
>
> **Q2: The experiment details of the simulated-robot task**
>
> We apologize for not having made the experimental details clear. The task is that **the robot aims to push a box from its initial point to the right side of the table**. With the box set at a base position and the robot arm set at a random position with Gaussian distribution, we set the target at its right and use the L2 distance of the box center and target position. We have made that clear in the revision of our paper.
>
> **Q3: What is the reason for not using a mixture model, e.g. GMM, to model this multi-modality demonstration behavior?**
>
> It is common for using a single Gaussian distribution policy to model behavior in Imitation Learning/Reinforcement Learning. Modeling it in GMM will cause tough issues. For example, when **using TRPO/PPO to optimize policy, there is no closed-form solution for KL-divergence constraint between two GMM to perform a trust region constraint**.
>
> Nevertheless, we find a way to implement the final policy as a GMM model by approximating the KL-divergence according to \[A, B\]:
>
> > If two Gaussian mixtures $f, g$ : $
> > f=\sum \alpha_{i} f_{i}$ and $g=\sum \beta_{i} g_{i}$ with weights $\alpha=\left(\alpha_{1}, \cdots, \alpha_{m}\right)$ and $ \beta=\left(\beta_{1}, \cdots, \beta_{m}\right)$, then their KL-divergence can be bound by: $K L(f \| g) \leq K L(\alpha \| \beta)+\sum \alpha_{i} K L\left(f_{i} \| g_{i}\right)$
>
> After learning transferability, we model our final policy as a GMM model and obtain the following result:
>
>
> |  Baseline  | Contrastive-Clustering (Ours) | GMM-policy  |
> | :--------: | :---------------------------: | :---------: |
> | -576 (±49) |         **-71** (±66)         | -388 (±484) |
>
> The low performance along with a high variance of a GMM policy verifies that due to optimization difficulties, **a GMM model cannot model multi-modal behavior accurately even after selecting those transferable demonstrations, let alone dealing with the unselected multi-modal dataset**. So We didn't choose to use a mixture model to model this multi-modal behavior, but this could be another interesting topic to work on.
>
> [A] Approximating the Kullback Leibler Divergence Between Gaussian Mixture Models. (2007)
>
> [B] Lower and upper bounds for approximation of the Kullback-Leibler divergence between Gaussian mixture models. (2012)
>
> **Q4: How to compute the transferability of the other 3 methods?**
>
> We compute transferability for the other 3 methods as they originally do (and the only difference would be on a mixed source of demonstrations), listed as follows:
>
> * f-MDP [C]: $w(\xi)=\exp \left(\frac{-\sum_{t=1}^{N}\left(\gamma^{f}\right)^{t} f_{\text {dis }}\left(s_{t}, s_{t}^{d}\right)-C}{\sigma}\right)$. Here, the parameter $C$ and $\sigma$ are constants that control scale and $\gamma$ the discount factor, $f_{dis}$ indicates the L2 distance.
>
> * ID & ID-GAIL [D]: $w_{f}(\xi)= \begin{cases}1 & F\left(\xi, \xi^{\prime}\right)<d_{\min } \\\\ 1-\frac{F\left(\xi, \xi^{\prime}\right)-d_{\min }}{d_{\max }-d_{\min }} & d_{\min } \leq F\left(\xi, \xi^{\prime}\right) \leq d_{\max } \\\\ 0 & F\left(\xi, \xi^{\prime}\right)>d_{\max }\end{cases}$. Here, $\xi^{\prime}$ is a corresponding trajectory to $\xi$ generated by the inverse dynamics $f_{\mathrm{id}}$. We choose $F\left(\xi, \xi^{\prime}\right)$ to be L2 as a sequence metric between the two trajectories.
>
> In our implementation of experiments, we use these weights as sampling weight $p_w\left(s_{t}^{d}, s_{t+1}^{d}\right)=\frac{w\left(s_{t}^{d}, s_{t+1}^{d}\right)}{\sum_{\left(s_{t^{\prime}}^{d}, s_{t^{\prime}+1}^{d}\right) \in \Xi} w\left(s_{t^{\prime}}^{d}, s_{t^{\prime}+1}^{d}\right)}$, as in $\underline{\text{Eq. (7) of main paper}}$. For more details, we refer to the reviewer for original papers of our baselines.
>
> [C] Z. Cao, Y. Hao, M. Li, and D. Sadigh. Learning feasibility to imitate demonstrators with different dynamics. In CoRL, 2021.
>
> [D] Z. Cao and D. Sadigh. Learning from imperfect demonstrations from agents with varying dynamics. IEEE Robotics and Automation Letters (RA-L), 2021

---

> > ### Author Response · Authors · 2022-08-24
> > **Response to Reviewer Zz2p (Part 2)**
> >
> > **Q5: Where did the demonstrations come from? From RL agent or some human input or hard-coded trajectories?**
> >
> > We apologize for this negligence in mentioning the source of our demonstrations. For MuJoCo environments, we train the experts with the TRPO algorithm to generate demonstrations, while in the Driving and the Simulated Franka Panda environments, we make demonstrations by handmade rules. We have made that clear in the revised version.
> >
> > **Q6: The selection and impact of the number of clusters  K**
> >
> > **(1)** K is unknown to us but we can estimate it empirically (by dimension reduction and then visualizing trajectories, for example). Like we do in the paper, we visualize trajectories under a MuJoCo setting with t-SNE and found approximately 4 modes, so as an approximation, we set K to 5 and the result turns out good. If one wants to get an optimal K, he may need to grid-search it around this approximation, but often an approximation is good enough. Note that the number of 'modes' has no direct relationship with the number of source domains, especially in a real-world scenario: data can be collected every day, and each day can be seen as a source, but they may all fall into a certain number of modes, i.e. the number of modes is not likely to increase unlimitedly.
> >
> > **(2)** The effect of K=2, 3, 5, ..., 20 (conducted on Hopper) are presented below which is already in the $\underline{\text{Section C.1 of supplementary material}}$,
> >
> > |     K=2      |     K=3     |     K=5     |    K=10     |    K=15     |    K=20     |
> > | :----------: | :---------: | :---------: | :---------: | :---------: | :---------: |
> > | 2613 (±1170) | 2685 (±475) | 3153 (±508) | 3130 (±121) | 3157 (±194) | 3041 (±245) |
> >
> > which shows that our algorithm is not very sensitive to K, and with larger K it achieves consistent performance because more clusters guarantee a clear separation between different modalities. Once the cluster number is enough to capture all the modalities, more clusters do not improve the performance.
> >
> > **(3)** After contrastive clustering, imitation learning on each cluster can be done parallelly, which can save time. Only a subset of demonstration can also be easier to fit than fitting the whole dataset, which also boosts learning efficiency. We admit it is a limitation that we cannot afford the huge computational cost if K increases into hundreds or thousands. Since the scalability of our current method is mainly limited by the computational cost of separate transferability learning on $K$ clusters, we hypothesize that pre-training or meta-training techniques may boost transferability learning efficiency, which is a promising future direction. We have added discussions about this in the revised Section 5 of main paper on limitations and future work.

---

> ### Author Response · Authors · 2022-08-27
> **Discussion period ends soon**
>
> Dear Reviewer,
>
> Thank you very much for your time and efforts in reviewing our paper. It is a kind reminder that **this is the last day of the Reviewer-author discussion**. Following your suggestion, we believe that we have made a great effort to provide all the experiments and clarifications that we can. Kindly let us know if we have addressed your concerns or if you have any additional comments.
>
> Looking forward to hearing from you. Thank you!

---

### Official Review · Reviewer_sqom · 2022-07-29

**Originality:** Good
**Technical Quality:** Fair
**Clarity Of Presentation:** Good
**Impact:** 3

**Recommendation:**

Weak Accept: I recommend accepting the paper, but will not argue for my recommendation if the majority of other reviewers have a different opinion.

**Summary:**

This paper presents a new Imitation Learning method designed to address action-space mismatch between demonstrator and learner. The action-space mismatch assumption manifests in the assumption that no expert actions are available, and also implies that the underlying dynamics of the demonstrations $p(s' | s, a_{\text{expert}})$ does not match the dynamics of the learner $p(s'|s, a_{\text{learner}})$. This is termed 'OOD-IL' (Out-of-dynamics IL). The proposed method uses contrastive learning and clustering to learn a set of discriminators and policies in a GAIL framework, which is designed to address potential multimodality in the training data. The resulting set of discriminators are used to form a joint distribution over state transitions that is intended to capture whether a particular state transition is feasible. This joint distribution is then used as the sampling distribution for performing a final stage of imitation learning in an 'OOD' (out-of-dynamics) environment.

**Issues:**

See the weaknesses section, in particular, W1. Based on my current understanding, following the protocol suggested in the experiments for W1 would be sufficient to address W1, if it indeed turns out to confirm that multimodality is the important key factor. However, it could turn out to be the case that these experiments do not show the multimodality to be important, in which case, I think the results would necessitate further explanation and the paper significantly reworked around a different hypothesis that is consistent with the experiments.

**Quality Of The Limitations Section:**

Additional details required

**Reviewer Expertise:**

4: The reviewer is confident but not absolutely certain that the evaluation is correct

**Robotics Focus:**

Highly relevant to robotics but no hardware experiments

**Strengths And Weaknesses:**

### Strengths
- The OOD-IL regmie is certainly of interest, as successful methods in this of relaxation of typical imitation learning would enable a much broader set of training data to potentially be used for imitation learning.
- The proposed method appears to work better than prior work in this regime.
- Ablations confirm the efficacy of the clustering component and the transfer component on one environment
- A sweep of the clustering hyperparameter (in the appendix) further confirms the efficacy of the clustering component

### Weaknesses
- W1 (main weakness): The main experimental flaw is that there seems to no evidence presented in support of the claim that the multimodality of the demonstrations is the main factor behind the performance difference between the proposed approach and other approaches. This is an important weakness because it is part of the paper's central claims. E.g. L48-50 "The multimodal demonstrations make it impossible to learn a deterministic policy from the f-MDP. Furthermore, f-MDP suffers from slow and inaccurate optimization difficulties due to a step-by-step optimization procedure"; L124-125 "This multimodal distribution of the demonstrations makes prior works ineffective. In this paper, we simultaneously remove the interference from the multimodal distribution and learn transferability to mitigate negative impact of non-transferable demonstrations." I expand on W1 below.

- W2: Discussion of prior methods that also perform clustering by contrastive learning are missing. I suppose the idea is natural, but the current paper would be strengthened by drawing connections to this line of prior work and discussing their significance to the current work. These papers look related [A, B, C].

- W3: I doubt the following claim. L181 claims that "we can embed our transferability into any imitation learning algorithm." More explanation is needed for why this is the case. For instance, how could one use this sampling distribution to perform BC? The expert actions at samples from $p_w$ are not necessarily available. Therefore, BC is a counterexample to refute this claim on L181. Either the claim needs to be removed or refined (inclusion of a refined explanation and/or evidence). The regime studied in the paper is particuarly one in which the expert actions for the robot are not available, so it seems to rule out any imitation learning algorithm that relies on these (i.e. we can only use state-based imitation learning algorithms). If that's the conclusion, it would be good to include in the appendix (and reference in the main text) an example objective function of a different imitation learning algorithm making use of $p_w$.

- W4: The limitations section needs work. It doesn't actually discuss limitations of the proposed method in the experimental settings investigated, which suggests that it 'solves' them.

**More details on W1**:
Because there are other differences between the proposed approach and prior works, a direct comparison between them does not necessarily identify the multimodality of the demonstrations as the source of the performance difference. Therfore, Figs 4, 5b, and 6b, -- the main analyis -- do not constitute conclusive evidence in support of the claim. Fig 7 contains an ablation of the clustering component, and the clustering component allows for more policies to be learned to fit the original (pre-transfer) data. However, even with the clustering component removed, because the policies are stochastic, they still have the capacity to fit multimodal trajectory distributions. Therefore, to me it seems there is no evidence presented in support of the main claim.

Here are my suggestions for experiments to run to collect this evidence:
- (1) Re-run the ablations in Fig 7 and also include ablations of the proposed method for the other environments, except with deterministic policies. This will force each policy to be unimodal, and therefore *only* the clustering (inclusion of multiple policies) would allow for fitting of multimodal data. Comparing the results of the four variants: {with clustering, without clustering} X {with deterministic policy, with stochastic policy} is the key idea here.
- (2) Confirm that the performance benefits of fitting multi-modal data go away when the data is made unimodal. This could be done by modifying the Driving environment so that only unimodal demonstrations and task. Then, there should be little-to-no performance difference between the ablations.

#### Minor weaknesses
- MW1: S4.4 and Fig7 are unclear -- what is the environment used for these experiments? I'm guessing it's the driving environment (based on the similarities of the y-axes in Fig5b and Fig7), but this needs to be made unambiguous.
- MW2: L202 "Hooper" -> Hopper

### References
- [A] "Deep Robust Clustering by Contrastive Learning" https://arxiv.org/pdf/2008.03030.pdf
- [B] "Contrastive Clustering" https://ojs.aaai.org/index.php/AAAI/article/download/17037/16844
- [C] "Supporting Clustering with Contrastive Learning" https://arxiv.org/pdf/2103.12953.pdf


**Summary Of Recommendation:**

The regime is interesting, and the method is interesting, yet the central claim is lacking evidence (see W1). While I think the work has potential for impact, this lacking evidence makes the paper technically lacking, and therefore I do not recommend acceptance of the paper in its current state.

---

> ### Author Response · Authors · 2022-08-24
> **Response to Reviewer sqom (Part 1)**
>
> Many thanks to Reviewer sqom for providing an insightful review and valuable comments.
>
> **Q1: Main weakness: identifying the multimodality of the demonstrations as the source of the performance difference.**
>
> We appreciate the suggestion by the reviewer of demonstrating the significance of the multimodality of the demonstrations.
>
> **(1) Comparison of deterministic policy and stochastic policy.** In our work, we model the policy output as a **uni-modal Gaussian distribution** in all environments, which has a variance around the mean/mode to fit the commonly-used log-likelihood loss for continuous  action but cannot fit multimodal trajectory distributions and thus can be seen as deterministic.  It is common for using a single Gaussian distribution policy to model behavior in Imitation Learning/Reinforcement Learning and only such a deterministic policy is considered in our original paper.
>
> So the first ablation experiment suggested by the reviewer considering using a deterministic policy is already done. **Using a stochastic policy suggests modeling it with a mixture of models like the Gaussian Mixture of Models, which however will cause tough issues**. For example, when using TRPO/PPO to optimize policy, there is no closed-form solution for KL-divergence constraint between two GMM to perform a trust region constraint.
>
> Nevertheless, we find a way to implement the final policy as a GMM model by approximating the KL-divergence according to \[A, B\]:
>
> > If two Gaussian mixtures $f, g$ : $
> > f=\sum \alpha_{i} f_{i}$ and $g=\sum \beta_{i} g_{i}$ with weights $\alpha=\left(\alpha_{1}, \cdots, \alpha_{m}\right)$ and $ \beta=\left(\beta_{1}, \cdots, \beta_{m}\right)$, then their KL-divergence can be bound by: $K L(f \| g) \leq K L(\alpha \| \beta)+\sum \alpha_{i} K L\left(f_{i} \| g_{i}\right)$
>
> After learning transferability, we model our final policy as a GMM model and obtain the following result in the Driving environment:
>
>
> |  Baseline  | Contrastive-Clustering (Ours) | GMM-policy  |
> | :--------: | :---------------------------: | :---------: |
> | -576 (±49) |         **-71** (±66)         | -388 (±484) |
>
> The low performance along with a high variance of a GMM policy verifies that due to optimization difficulties, a GMM model shows poor performance for fitting multimodal data accurately even after selecting those transferable demonstrations. So though the claim of the reviewer : 'because the policies are stochastic, they still have the capacity to fit multimodal trajectory distributions' seems natural, **it is challenging to model a stochastic policy in implementation when dealing with a multi-modal dataset**. So We didn't choose to use a stochastic policy and instead turned to an auxiliary clustering step. But this could be another interesting problem to work on.
>
> \[A\] Approximating the Kullback Leibler Divergence Between Gaussian Mixture Models. (2007)
>
> \[B\] Lower and upper bounds for approximation of the Kullback-Leibler divergence between Gaussian mixture models. (2012)
>
> **(2) Ablation study on the uni-modal environment.** We further confirm that with only unimodal demonstrations and tasks, there should be little-to-no performance difference between the ablations. The result is presented below:
>
> |                     |  Random Policy  | Baseline   | Ours w/o Cluster |      Ours      | $\Delta$ (Impact of Clustering) |
> | ------------------- | :-------------: | ---------- | :--------------: | :------------: | :-----------------------------: |
> | Driving-multi-modal | $\approx$ -1000 | -576 (±49) |    -494 (±68)    | **-71** (±66)  |          423 (**Big**)          |
> | Driving-uni-modal   | $\approx$ -1000 | -592 (±77) |    -554 (±77)    | **-519** (±78) |         35 (**Small**)          |
>
> Here are the experimental details. In the Driving environment, we constructed the uni-modal environment by **collecting from 3 sources, each containing no overlapping trajectories to the other sources**. The first source of expert demonstrations is initialized at the left part, then go straight forward; the second source of expert demonstrations is also initialized at the left part but makes a slight turn towards the middle part and goes straight forward; the third source of expert demonstrations is initialized at the middle part and go straight forward. While demonstrations from the latter 2 sources are transferable to the target environment, the first 1 source contains no transferable demonstrations. We achieve this by placing an obstacle on the left part of the target environment. For sources and the target environments, creating obstacles at different places is how we create different dynamics for this Driving-uni-modal setting.
>
> The ablation on Driving-uni-modal shows that, **after removing the multi-modality of demonstrations, w/ and w/o clustering shows NO significant difference**, while we emphasize the necessity for clustering in a multi-modal dataset, which almost is always the case in the real world.

---

> > ### Author Response · Authors · 2022-08-24
> > **Response to Reviewer sqom (Part 2)**
> >
> > (Q1 continued)
> >
> > **(3) Claims clarification.** Some expressions might not be clear enough to support the claim that the multimodality of the demonstrations is the main factor behind the performance difference between the proposed approach and other approaches. We clarify this claim as follows:
> >
> > L48-50 "*The multimodal demonstrations make it impossible to learn a deterministic policy from the f-MDP. Furthermore, f-MDP suffers from slow and inaccurate optimization difficulties due to a step-by-step optimization procedure*": This problem faced by f-MDP is observed during our first few attempts to reproduce its results on multiple constructions of datasets. Due to the mutual influence of multimodal distributions, along with the difficulty and instability of the RL algorithm itself, f-MDP approach often fails to learn an agent with a deterministic policy that can mimic those transferable demonstrations in a feasibility-MDP process. So **first** we mitigate this problem of negative mutual influence of multimodal demonstrations by clustering demonstrations into uni-mode. This conclusion is drawn according to the ablation study of 'Ours w/o Cluster' on both multi-modal and uni-modal datasets. **Then**, the optimization problem further can be addressed by using the output of the discriminator to distinguish transferable demonstrations from non-transferable ones, because it measures the distance of two distributions instead of the L2 distance of two instances as f-MDP does. **These two designs we have made are our key contributions of us compared with other baseline approaches**.
> >
> > **Q2**: **Discussion of prior methods that also perform clustering by contrastive learning**
> >
> > We appreciate your suggestion about drawing connections to the clustering-by-contrastive-learning line of prior works and discussing their significance to the current work. We have included discussions about these related paper in Section 2 of the revised paper.
> >
> > > While prior works either focus on visual representations in the form of a single fixed-length vector [A, B] or rely on augmentations with a heavy computational burden [C], our Sequence-based Contrastive Clustering is **unique in handling trajectory data with various sequential lengths, by utilizing simple and efficient down-sampling to construct positive pairs of contrastive learning.**
> >
> > [A] H. Zhong, C. Chen, Z. Jin, and X.-S. Hua. Deep robust clustering by contrastive learning.
> >
> > [B] Li, P. Hu, Z. Liu, D. Peng, J. T. Zhou, and X. Peng. Contrastive clustering. In AAAI, 2021.
> >
> > [C] D. Zhang et al. Supporting clustering with contrastive learning. In NAACL, 2021.
> >
> > **Q3**: **Clarification of the claim in L181 that "we can embed our transferability into any imitation learning algorithm."**
> >
> > We thank the reviewer for reminding us that this claim is not rigorous enough. It should be "we can embed our transferability into any **imitation-learning-from-observations** algorithm," even including BC [D] if we embed an inverse dynamics model to learn the action. Other algorithms like AIRL-from-observations can also be used for our setting apart from GAIL. We have rephrased that claim in the revised version.
> >
> > [D] F. Torabi, G. Warnell, and P. Stone. Behavioral cloning from observation. In IJCAI, 2018.
> >
> > **Q4: Limitation section**.
> >
> > We appreciate the suggestion for a rethinking of our limitation section. We have enriched our discussion on limitations and future work, regarding learning efficiency, safety constraints on a real system, and the quality of demonstrations in Section 5 of the revised paper. Please check our revised paper for this updated part.
> >
> > **Other minor issues:**  We fixed the typos and make sure the illustrations for figures are clear. Thanks for the reminder.

---

> ### Author Response · Authors · 2022-08-27
> **Discussion period ends soon**
>
> Dear Reviewer,
>
> Thank you very much for your time and efforts in reviewing our paper.
> It is a kind reminder that **this is the last day of the Reviewer-author discussion**. Following your suggestion, we believe that we have made a great effort to provide all the experiments and clarifications that we can. Kindly let us know if we have addressed your concerns or if you have any additional comments.
>
> Looking forward to hearing from you. Thank you!

---

### Official Review · Reviewer_M2Dq · 2022-07-31

**Originality:** Good
**Technical Quality:** Good
**Clarity Of Presentation:** Good
**Impact:** 3

**Recommendation:**

Weak Reject: I recommend rejecting the paper, but will not argue for my recommendation if the majority of other reviewers have a different opinion.

**Summary:**

The paper considers the problem of imitation learning from out-of-dynamics demonstrations. Since the demonstrators may have different dynamics than the imitator, these demonstrations may not be useful to the learner, if not detrimental. The paper follows existing work and uses a 2-stage procedure. It first learns a feasibility/transferability score for the demonstrations, and then uses an imitation learning algorithm on weighted data. The paper addresses a major limitation of f-MDP, the state-of-the-art approach, which is not able to deal with multimodal demonstrations. The proposed approach first clusters the demonstrations, then learns a policy for each cluster. Additionally, it uses a transferability metric that is easier to compute than f-MDP. The proposed approach is evaluated on various simulated tasks with different physics backends such as mujoco and pybullet.

**Issues:**

See **Strengths and Weaknesses**.

**Quality Of The Limitations Section:**

Additional details required

**Reviewer Expertise:**

4: The reviewer is confident but not absolutely certain that the evaluation is correct

**Robotics Focus:**

Relevant but unlikely to deploy to hardware in near future

**Strengths And Weaknesses:**

### Strengths
- **Motivation.** The research problem is well motivated. The challenges of the research problem, as well as the limitations of existing methods are presented clearly.
- **Organization.** The paper is in general well organized. The technical approach is easy to follow, and seems technically sound.
- **Literature survey.** The authors did a thorough review of related literature. The authors clearly present the limitations of existing work, and how the proposed work is able to address these limitations.
- **Ablation studies.** The authors present a series of ablation studies verifying the importance of each component, as well as demonstrating how each hyper-parameter is chosen.

### Weaknesses
- **Lack of real-robot results.** Although the proposed method is evaluated on different physics engines such as mujoco and bullet, it lacks experiments on real hardware. It seems like there are still considerable gaps between the simulation and real hardware experiments. First of all, the physics of real hardware could be very different from simulation. For example, in the videos on the website, the Franka robot and the box are often “sinking” into the table. Second, it is possible that demonstrations generated by RL policies in simulation differ from demonstrations that can be collected on the real hardware. Therefore, it would be nice if the authors can present results on real hardware, show that real-world demonstrations also have such multi-modal properties, and demonstrate that the proposed method outperforms baselines for such data.
- **Comparison with other clustering algorithms.** It would be nice if the authors can compare their results with simpler clustering algorithms, such as K-means. Although in the appendix, the authors claimed that L2 distance can’t be used on trajectories of different lengths, it seems like trajectory segments with fixed length are actually used according to the appendix.
- **Loss function.** In Equation (8), it seems like $p_w$ is also a function of the discriminator $D$. How did the authors compute the gradient of $D$ in this case? If the authors did not consider the gradient of $D$ from $p_w$, the authors may want to note that in the paper.
- **Final policy.** It seems like $K$ policies are trained by the proposed algorithm. Is the reported results from the mixture of $\pi_k$ or the best one?
- **Source of multimodal demonstrations.** It seems like that in a lot of the experiment setup, the demonstrations are multimodal because they come from different dynamics. It would be nice if the authors can compare their results with a baseline where a policy is learned for each demonstrator (with transferability but without clustering). This can significantly strengthen the paper by verifying the importance of the clustering step.
- **Data distribution for mujoco environments.** It seems that the composition of demonstrations is quite different among the mujoco environments. It is appreciated that the authors show results with different data composition for Hopper, but it would be nice if the authors show similar results for the other two environments.
- **HalfCheetah.** It seems like for HalfCheetah, all demonstrations are feasible, since $\gamma_f$ and $\gamma_b$ from each demonstrator is smaller than the imitator. Is that true?
- **Ablation study with trans.** It is appreciated that the author presented the ablation study. It would be nice if the authors also compare their results with the one that uses clustering but not the transferability step. It can help validate the importance of the transferability step.
- **Common $\rho_0$.** It seems like the authors assume that all of the MDPs share the same initial state distribution, why is such assumption necessary? By the way, It seems like $\rho_0$ is actually defined in the MDP paragraph.
- **Feature extractor F.** Feature extractor $F$ first appears in Equation (1), but is not described until line 149. The authors are also advised to provide more details on the structure of $F$ that was used in the experiments.
- **Typo?** In Equation (5), should it be $(s_t^d, s_{t+1}^d)\sim\Xi_k$ under the first expectation?
- **Transferable demonstrations.** In appendix C.3, the authors mentioned “equal number of transferable and non-transferable demonstrations”. How do the authors know which demonstrations are transferable, which are not?

**Summary Of Recommendation:**

The paper addresses an interesting research problem that is well-motivated. The authors were able to clearly explain the contribution of the proposed work. Although the proposed method seems technically sound, the contribution seems a little incremental. The main weakness of the paper comes from the experiment section. I am concerned about the paper because no real robot experiments are presented, and some additional results are needed to support the claims.

With that said, I am happy to increase my score if the authors present more experimental results during the rebuttal period.

---

> ### Author Response · Authors · 2022-08-24
> **Response to Reviewer M2Dq (Part 1)**
>
> We would like to sincerely thank Reviewer M2dq for providing a detailed review and insightful questions.
>
> **Q1:** **Lack of real-robot results**
>
> We apologize for the lack of real-robot experiments. Due to the COVID-19 pandemic, the robot in our lab is inaccessible to us, so we regret that we cannot conduct a real-robot experiment to verify the effectiveness of our method.
>
> However, we have rerun our simulated-robot experiments after fixing the unreal effect of the box sinking into the table (by making the simulator detect collision more accurately). Check the updated video on our anonymous [website](https://sites.google.com/view/oodil). We have also updated Figure 6(b) in the uploaded revision.
>
> | Naive GAIL |   f-MDP    |    ID     |  ID-GAIL   |      Ours      |
> | :--------: | :--------: | :-------: | :--------: | :------------: |
> |  34 (±76)  | 237 (±302) | 78 (±256) | 213 (±242) | **456** (±184) |
>
> Final performances for imitation learning are presented above (averaged over 3 seeds). We found that the results remain consistent with those presented in the original paper, which shows that directly applying GAIL or dealing with other baselines introduces a low performance, while the proposed method makes the new robot successfully learn from transferable demonstrations. Though there's no real-robot experiment, we assume these supplementary experiments on a simulated robot would roughly reflect the behavior in a real robot system.
>
> **Q2:** **Comparison with other clustering algorithms**
>
> We conducted the following experiments on the Driving environment by **K-Means clustering** with the number of clusters $K$=$10$ as in our method. Specifically, we down-sampled each trajectory with a fixed stride to uniformly generate a fixed-length subsample, and applied the K-means algorithm directly to these sub-trajectories and therefore assign each trajectory to a cluster. Then, on each of these K clusters, we learn the transferability respectively. The result of using transferability generated by K-Means-Clustering to do the final GAIL is presented below:
>
> |  Baseline  | K-Means-Clustering | Contrastive-Clustering (Ours) |
> | :--------: | :----------------: | :---------------------------: |
> | -576 (±49) |    -367 (±159)     |         **-71** (±66)         |
>
> We observed that the lacking of a contrastive learning step may cause difficulty in obtaining a high-quality unimodal cluster, which is essential for learning an accurate transferability measurement, and further cause a final performance drop. One way our contrastive clustering method is superior to K-means is that **performing K-means on uniformly random-sampled sub-trajectories may introduce high variance into the clustering results**, while our method, which makes different subsamples of the same trajectory as positive pairs and minimizes their distance in the hidden representation space, can mitigate such instability. Also, the extracted representations are used for clustering, so it is beneficial if they are learned with the clustering step in a coherent manner. We have added the implementation details and results in Section C.5 of the revised supplementary material.
>
> **Q3:  Using the source of multimodal demonstrations as another baseline for 'w/o clustering' ablation**
>
> We added an ablation study according to the reviewer's suggestion to verify the significance of the clustering step, by assigning cluster ID to each trajectory according to their sources. That is, in the ablation study on the Driving environment, we consider 3 sources as 3 clusters and then learn the transferability of each cluster respectively. The final performance is presented below:
>
> |  Baseline  | Source as Cluster | Contrastive-Clustering (Ours) |
> | :--------: | :---------------: | :---------------------------: |
> | -576 (±49) |    -457 (±66)     |         **-71** (±66)         |
>
> We concluded that **using the source as cluster ID does not bring promising results**. It is because the concept of each 'source' is not directly related to the concept of 'mode', i.e. there can be multimodal demonstrations even in one source. It depends on how we collect our dataset and the 'ground-truth' of mode distribution is unknown, especially in realistic applications where a source usually consists of a complex composition of different factors, such as periods of time, weather or traffic conditions. Therefore, knowing the source label of each source of multimodal demonstrations does NOT indicate knowing one domain/one mode, making clustering an indispensable process for separating different modes of demonstrations and constructing accurate transferability.

---

> > ### Author Response · Authors · 2022-08-24
> > **Response to Reviewer M2Dq (Part 2)**
> >
> > **Q4: Result with different data composition for MuJoCo environments**
> >
> > To address the reviewer's concern and better verify that our algorithm can handle various demonstration setups, we additionally show results with different data compositions for Walker2d as well as HalfCheetah like we already did for Hopper in Appendix.C.3. The compositions of demonstrations are set the same as in original paper, which is (i) (1, 0.9), (ii) (0.9, 1), (iii) (1, 0.05), (iv) (0.05, 1), with setting (·, ·) as the force of the front leg and the back leg for HalfCheetah, and setting friction to (i) 24.8, (ii) 9.9, (iii) 3.9, (iv) 1.1 for the Walker2d environment.
> >
> > The result, which has also been added to Section C.3 of the revised supplementary material, shows that our algorithm is robust and presents high performance consistently under different compositions of demonstrations, while other baselines deteriorate extremely as the ratio of transferable demonstrations goes down.
> >
> > Walker2d:
> >
> > | Composition | Naive GAIL |   f-MDP    |     ID      |   ID-GAIL   |      Ours       |
> > | :---------: | :--------: | :--------: | :---------: | :---------: | :-------------: |
> > |   1:2:2:2   | 318 (±290) | 283 (±190) | 1688 (±218) | 1703 (±175) | **2077** (±216) |
> > |   1:2:5:5   | 288 (±72)  | 249 (±37)  |  345 (±92)  |  328 (±39)  | **1731** (±73)  |
> > |  1:2:10:10  | 327 (±65)  | 213 (±48)  |  311 (±29)  |  349 (±30)  | **1664** (±166) |
> > |  1:2:20:20  | 287 (±74)  | 339 (±131) |  345 (±73)  |  320 (±64)  | **1629** (±87)  |
> >
> > Due to the time limit, we haven't finished the experiments on HalfCheetah environment, but we promise to update it in the following days for our final version.
> >
> > **Q5: How we define "transferable demonstrations"**
> >
> > Here, **the concept of 'Transferable' and 'Non-transferable' is unknown to us unless applying imitation learning directly to the demonstrations, according to the final imitation results**. For a better presentation of our experiments, we assume we ideally know whether the demonstrations are transferable (meaning the dynamics of the source environment are set close to the target and source demonstrations are easy to imitate) in the experimental setup to show how the compositions are different.
> >
> > Specifically, in the HalfCheetah environment, different joint motor forces are discounted to different levels in each source. We still assume that the closer the dynamics are set, the more likely those demonstrations generated in such source dynamics are transferable to the target environment.  The scaling of gear doesn't necessarily mean that scaling action to the same extent would generate the same trajectory, as the underlying physics model of an agent is always very complex and most of the time the relationship between the gear force and the observed behavior is nonlinear, e.g. the interaction force between different joints need to be considered. Therefore, we adopted the assumption that setting the dynamics of the source environment close to the target will make the demonstrations more transferable. Thus, **even the target environment is set with $\gamma_f,\ \gamma_b$ not discounted, those trajectories generated in a largely discounted environment still are infeasible to the target environment.**
> >
> > **Q6: Ablation study with Trans**
> >
> > The reviewer suggested that 'It would be nice if the authors also compare their results with the one that uses clustering but not the transferability step'.
> >
> > However, the clustering step serves for the transferring step (i.e. to derive an accurate transferability after splitting demonstrations into different modes), so actually, **the ablation for 'w/ Cluster and w/o Trans' is equivalent to Naive GAIL**, which is already presented in the main paper. Thus, we cannot add more ablations as the reviewer suggests.
> >
> > So, we provide **the best performance of $\pi_k$ learned for each cluster** which may be seen as additional results only using clustering and does not consider the transferability step. However, the $\pi_k$ provided within each cluster may not fully learn optimal actions on all states, as it is only trained with a subset of demonstrations, so if imitation learning is conducted on each cluster without integrating them, the policy we derive can be incomplete. **The best performance of policies $\pi_k$ is reported below and is observed inferior to the final policy**. Therefore, transferring all possible transferable demonstrations across all clusters is necessary.
> >
> > |                 | Best $\pi_k$ w/ Cluster |  Final $\pi$  |
> > | :-------------: | :-----------------------: | :-------------: |
> > |     Hopper      |        2461 (±837)        | **3010** (±45)  |
> > |    Walker2d     |        1606 (±102)        | **2077** (±216) |
> > |     Driving     |        -236 (±90)         |  **-71** (±66)  |
> > | Simulated robot |         278 (±89)         | **456** (±184)  |

---

> > > ### Author Response · Authors · 2022-08-24
> > > **Response to Reviewer M2Dq (Part 3)**
> > >
> > > **Q7: The problem of common** $\rho_0$
> > >
> > > We assume that the source and target MDP share the same initial state distribution $\rho_0$ as we commonly do in Imitation Learning in Varying Dynamics regimes, like [A] [B] (also cited by the paper). But we assume that it may be possible to relax this assumption because we can also filter out those demonstrations with initial states not shared with the target environment, by implementing our transferability measurement. We may explore this idea in future work.
> > >
> > > [A] Z. Cao, Y. Hao, M. Li, and D. Sadigh. Learning feasibility to imitate demonstrators with different dynamics. In CoRL, 2021.
> > >
> > > \[B\] Z. Cao and D. Sadigh. Learning from imperfect demonstrations from agents with varying dynamics. IEEE Robotics and Automation Letters (RA-L), 2021
> > >
> > > **Other minor issues:**
> > >
> > > - **Loss function**: In the final phase, $p_w$ is used as a fixed ratio and has no gradient. We have made that clear in the revised version.
> > >
> > > - **Final policy**: As we have described in the last paragraph of $\underline{\text{Section 3.2 of main text}}$, after training K policies, we **trained another policy with the re-weighted sampling of all demonstrations which gives the final result**.
> > >
> > > - **Feature extractor**: Feature extractor $F$ is described in $\underline{\text{Line 140 of the original main paper}}$ and implementation details are in $\underline{\text{Section B of supplementary material}}$.
> > >
> > > Other typos are also fixed. Thanks for the reminder.

---

> > > > ### Comment · Reviewer_M2Dq · 2022-08-27
> > > > **Thank you for your response**
> > > >
> > > > Thank you for your response! I appreciate that the authors conduct additional experiments of 1) different data distribution, 2) comparing $\pi_k$ and the final policy, as well as 3) comparing the proposed method with K-means. These results are convincing and cleared up a lot of my concerns. It is also great that the authors make the simulation more realistic. I will consider increasing my score in the final phase.

---

> > > > > ### Author Response · Authors · 2022-08-28
> > > > > **Thanks for the Response of Reviewer M2Dq**
> > > > >
> > > > > Dear reviewer,
> > > > >
> > > > > Thanks again for your dedication to reviewing our paper and your appreciation of our feedback. Your detailed suggestions help us a lot in improving our work.
> > > > >
> > > > > All the best,
> > > > >
> > > > > Authors.

---

> ### Author Response · Authors · 2022-08-25
> **Update: Addtional results of different data compositions for HalfCheetah**
>
> Dear Reviewer,
>
> We added the experiments of different data compositions in HalfCheetah. Please check the following results:
>
> ----
>
> | Composition | Naive GAIL  |    f-MDP    |     ID      |   ID-GAIL   | Ours            |
> | :---------: | :---------: | :---------: | :---------: | :---------: | --------------- |
> |   2:2:1:1   | 2389 (±897) | 404 (±246)  | 2031 (±312) | 2210 (±86)  | **3008** (±117) |
> |   2:2:2:2   | 2882 (±84)  | 247 (±308)  | 2126 (±110) | 2067 (±86)  | **2997** (±209) |
> |   2:2:5:5   | 2201(±502)  | 1613 (±409) | -327 (±119) | 1273 (±546) | **3249** (±134) |
> |  2:2:10:10  | 2367 (±897) | 389 (±232)  | 1808 (±146) | 1315 (±414) | **2981** (±71)  |
> |             |             |             |             |             |                 |
>
> We concluded from the results that the proposed method achieves promising average returns under different scenarios stably, with expert demonstrations that can approximately achieve 3200 average return. This conclusion is in consistence with that of **Q4** on the other 2 MuJoCo environments. We have added the results to our revised supplementary material in Section C.3.

---

> ### Author Response · Authors · 2022-08-27
> **Discussion period ends soon**
>
> Dear Reviewer,
>
> Thank you very much for your time and efforts in reviewing our paper. It is a kind reminder that **this is the last day of the Reviewer-author discussion**. Following your suggestion, we believe that we have made a great effort to provide all the experiments and clarifications that we can. Kindly let us know if we have addressed your concerns or if you have any additional comments.
>
> Looking forward to hearing from you. Thank you!

---

### Meta-Review · Area_Chair_B791 · 2022-08-13

**Recommendation:** Accept (Poster)
**Confidence:** 4

**Metareview:**

This paper studies an important problem of learning from multimodal demonstrations where multiple demonstrators have different dynamics than the imitator. This is relevant to the robotics community, though there were no real robotics experiments. The reviewers found the work to be original and well-motivated, easy to follow, and appreciated the inclusion of several ablation studies. The limitations of the approach were discussed but could be improved. Comparisons to different clustering methods and investigating the impact of the number of clusters would improve the paper. The authors are encouraged to address these concerns along with other concerns raised by the reviewers.

============

Thank you to the authors for addressing the authors concerns. I believe the paper makes enough contributions to warrant acceptance to the conference. One reviewer has a remaining concern of statements made within the paper as summarized here:

A deterministic policy can indeed model a multimodal trajectory distribution, as long as the dynamics distribution is stochastic, which can generally cause a policy to lead to different outcomes for repeated trials from the same initial state.

Furthermore, even if the modelled policy is Gaussian, it cannot be seen as deterministic. A Gaussian distribution is never deterministic, except for the degenerate, 0-variance case.

Please address these concerns in the text of the final paper.


**Best Paper Nomination:**

No